# Interspecies Blastocyst Complementation and the Genesis of Chimeric Solid Human Organs

**DOI:** 10.3390/genes16020215

**Published:** 2025-02-12

**Authors:** Elena Bigliardi, Anala V. Shetty, Walter C. Low, Clifford J. Steer

**Affiliations:** 1Department of Neurosurgery, University of Minnesota, Minneapolis, MN 55455, USA; lowwalt@umn.edu; 2Molecular, Cellular, Developmental Biology, and Genetics Graduate Program, University of Minnesota, Minneapolis, MN 55455, USA; shett098@umn.edu; 3Stem Cell Institute, University of Minnesota, Minneapolis, MN 55455, USA; 4Division of Gastroenterology, Hepatology and Nutrition, Department of Medicine, University of Minnesota, Minneapolis, MN 55455, USA

**Keywords:** blastocyst complementation, chimerism, genetic engineering, intra-/interspecies, pluripotent stem cells, porcine, transplantation, xenotransplantation

## Abstract

Solid organ transplantation remains a life-saving treatment for patients worldwide. Unfortunately, the supply of donor organs cannot meet the current need, making the search for alternative sources even more essential. Xenotransplantation using sophisticated genetic engineering techniques to delete and overexpress specific genes in the donor animal has been investigated as a possible option. However, the use of exogenous tissue presents another host of obstacles, particularly regarding organ rejection. Given these limitations, interspecies blastocyst complementation in combination with precise gene knockouts presents a unique, promising pathway for the transplant organ shortage. In recent years, great advancements have been made in the field, with encouraging results in producing a donor-derived organ in a chimeric host. That said, one of the major barriers to successful interspecies chimerism is the mismatch in the developmental stages of the donor and the host cells in the chimeric embryo. Another major barrier to successful chimerism is the mismatch in the developmental speeds between the donor and host cells in the chimeric embryos. This review outlines 19 studies in which blastocyst complementation was used to generate solid organs. In particular, the genesis of the liver, lung, kidney, pancreas, heart, thyroid, thymus and parathyroids was investigated. Of the 19 studies, 7 included an interspecies model. Of the 7, one was completed using human donor cells in a pig host, and all others were rat–mouse chimeras. While very promising results have been demonstrated, with great advancements in the field, several challenges continue to persist. In particular, successful chimerism, organ generation and donor contribution, synchronized donor–host development, as well as ethical concerns regarding human–animal chimeras remain important aspects that will need to be addressed in future research.

## 1. Introduction

In the United States alone, the total number of solid organ transplants has increased from 83 transplants per million to almost 140 transplants per million since 2000 [1]. Unfortunately, despite increasing numbers of deceased organ donors, it is estimated that 17 people still die each day while on the organ transplant waiting list [1,2]. The supply of living and deceased organ donors cannot cover the current requirements, making the search for alternatives even more important.

One such alternative involves the production of exogenic organs. Historically, this has been attempted with the xenotransplantation of solid organs. A more recent example of this was in 2022, whereby a gene-edited porcine heart was transplanted in a patient with severe heart failure and allowed for the longest recorded survival of 7 weeks following xenotransplantation [3]. Further advancements have opened the pathway to blastocyst complementation, an important technique that would allow for interspecies tissue and organ production.

The first requirement for successful blastocyst complementation involves the genetic modification of a host blastocyst. The host must undergo knockout or gene modification that leads to a developmental defect of the organ of choice and thereby opens a niche for wild-type stem cells to occupy and develop into the desired organ. Techniques used to achieve this often include CRISPR/Cas9 (clustered regularly interspaced short palindromic repeats/CRISPR-associated protein 9) or TALEN (transcription activator-like effector nucleases), as they have shown most success in targeted genomic modification. Donor ESCs (embryonic stem cells) or iPSCs (induced pluripotent stem cells) are then introduced and injected into the host blastocyst. During subsequent embryonic in vivo organ development, the organ of choice is primarily made up of donor cells, which can then be transplanted back into the donor (See Figure 1) [4].

First attempts at this were carried out in 1993, whereby ESCs (embryonic stem cells) were injected into RAG2 (recombination activating gene 2 protein)-deficient mice to generate chimeric B and T cells [5]. Subsequent studies have shown that interspecies chimerism could be possible. For example, Xiang et al. used two distant rodent species to demonstrate that the injection of donor ESCs into a host resulted in most organs containing donor-derived cells, with some tissues showing an up to 40% contribution [6]. This innovative technique offers a promising pathway to help with the continuing transplant organ shortage. As such, this review aims to discuss the advances made in blastocyst complementation for solid organs. A summary of the results from all 19 studies can be found under Table 1.

Xenotransplantation: The host (pig) is modified to reduce the risk of rejection. This includes the knockout of growth hormones (to reduce intrinsic host growth) and xenoantigens, in addition to the inactivation of PERVs (porcine endogenous retroviruses). Human transgenes are also expressed to improve their compatibility. The new organ is mostly made up of host cells, then transplanted into the human.

Blastocyst Complementation. The host, a pig embryo, for example, is first genetically modified, often using TALEN or CRISPR, for organ agenesis to produce an organ niche. Human donor pluripotent stem cells are then introduced in the host blastocyst, allowing for the development of a chimeric host with an organ of choice. The organ, which is mostly made up of donor–human cells, is then re-transplanted into the donor.

## 2. Liver

Per the CDC (Centers for Disease Control and Prevention), the number of deaths due to chronic liver disease surpassed 50,000 in 2022 [26]. For patients with chronic liver disease or liver failure, orthotopic liver transplantation remains a critical, life-saving form of treatment. In 2022, over 10,000 patients required liver transplantation in the United States, making it the second most frequent solid organ transplant [1]. The need for viable transplants, therefore, remains crucial for the survival of many patients. The potential of iPSCs in post-natal hosts has been evaluated. Several studies have used human iPSCs, which were induced to differentiate into functional hepatocyte-like cells. These cells were then successfully transplanted into injured mouse livers or livers in hepatic failure (usually induced by CCl4). The results from these studies show success in improving the survival or function (measured via increased albumin or decreased bilirubin and LDH) [27,28,29]. For example, in a more recent study by Takayama et al., mice injected with human iPSC-derived hepatocyte-like cells showed a 50% survival rate at 15 days compared to 0% in sham-mice. Furthermore, in a chronic liver failure model, transplanting the above-mentioned cells resulted in improved hepatic functionality, with significantly increased albumin levels and a decreased expression of fibrosis markers, thereby rescuing the mice from chronic liver injury [30]. Successes in this area and subsequent novel uses of human ESCs or iPSCs have shown that human organ development in a non-human host could be achieved via blastocyst complementation.

### 2.1. Hepatogenesis and Elimination of Hepatic Development

The liver is a complex organ. While certain resident cells, such as Kupffer, Ito or stromal cells, are of a mesodermal origin, the liver’s primary cells, i.e., hepatocytes and cholangiocytes, stem from the endoderm. From the endoderm, the foregut, midgut and hindgut are established. The ventral foregut is what eventually develops into the liver bud. Development from the liver bud into hepatoblasts requires inductive signaling from the neighboring cardiac mesoderm [31]. The final differentiation is, in part, determined by the localization of the hepatoblasts. Differentiation into cholangiocytes occurs in cells next to portal veins (the caudal section of the liver bud), while cells located in the parenchyma give rise to hepatocytes (the cranial section of the liver bud) [32]. Naturally, there are several transcription regulators that have been found to be crucial for the successful development of the liver, including FoxA1-3 (forkhead box protein), GATA4/6 (guanine-adenine-thymine-adenine), HNF1α/β (hepatocyte nuclear factor), HNF4α, HNF6, OC-2 (onecut), C/EBPα/β (CCAAT/enhancer-binding protein), Hhex (haematopoietically expressed homeobox) and Prox1 (prospero-related homeobox) [33,34]. Of those, Hhex, in particular, has shown great promise as a target for preventing hepatogenesis. Hhex is a homeobox gene and is critical for liver differentiation and hepatobiliary development, as its absence prevents successful migration to the pseudostratified epithelium and subsequent hepatoblast differentiation [35].

However, Hhex mutations have been shown to affect the development of several systems, including the hematopoietic system, vascular system, forebrain and thyroid [36,37,38]. As such, different targets have been investigated, including FAH (fumarylacetoacetate hydrolase). A deficiency in FAH has shown to lead to apoptosis and mutagenesis in hepatocytes due to the accumulation of the toxic metabolic fumarylacetoacetate, particularly in hepatocytes [39]. FAH-deficient mice have also been shown to successfully be repopulated with hepatocytes when injected with human hepatocytes [40].

### 2.2. Application of Blastocyst Complementation

While iPSCs have shown promising potential for several applications, including regenerative medicine, disease modeling and gene therapy for inherited liver diseases, blastocyst complementation for organ development has also shown great promise [34].

Espejel et al. showed early promise in an FAH-deficient murine model. Of the 40 FAH-deficient blastocysts injected with iPSCs, 6 demonstrated sufficient chimerism for survival without requiring subsequent treatment for liver failure. Liver biopsies from highly chimeric mice showed successful liver hepatocyte repopulation by post-natal day 70. Fully iPSC-derived hepatocytes demonstrated about 80% FAH expression when compared to wild-type hepatocytes. Furthermore, the highly chimeric mice demonstrated normal liver function tests up to day 300, with no significant difference in bilirubin, albumin, alkaline phosphatase and alanine aminotransferase levels when compared to wild-type mice [7].

Matsunari et al. further investigated the use of blastocyst complementation using a pig model. They demonstrated two key elements. Firstly, they showed that bi-allelic Hhex-knockout using TALEN in pigs successfully led to severe liver dysplasia and developmental retardation. Secondly, they demonstrated blastocyst complementation could be used to develop livers in a pig model. Initial blastocyst complementation produced 37 blastocysts; of those, only 4 were chimeric fetuses. While 3 of the chimeric fetuses continued to demonstrate high developmental retardation, one of the chimeric fetuses showed organogenesis similar to that of the wild-type fetuses. Further work with 95 chimeric, blastocyst-complemented fetuses resulted in 3 live fetuses with normal liver development at the time of cesarean section [8].

Ruiz-Estevez et al. similarly produced Hhex knockouts in pigs and mice using CRISPR/Cas9, with an absence of hepatogenesis in both species [9]. The knockout of Hhex was embryonically lethal around E10.5 for both species. Intra-species blastocyst complementation was then performed, with eGFP-labeled donor-derived cells (enhanced green fluorescent protein). Their study showed a very high contribution of donor-derived cells (evidenced by a positive eGFP signal) in several tissues, including the liver. In mice, 22/32 embryos demonstrated an eGFP-positive signal, with 50% surviving past E10.5. In pigs, only 2 embryos were recovered from 94 total complemented embryos. However, both these embryos demonstrated a strong eGFP signal, with visible liver tissue and entirely Hhex wild-type sequences in the liver.

Most recently, Simpson et al. demonstrated conditional Hhex knockout in pigs [10]. As discussed previously, the knockout of Hhex has widespread consequences outside of just liver development in the body, such as arrested forebrain development and changes in endothelial cell differentiation and lymphatic vessel formation. As such, a more targeted knockout would be of interest to minimize the off-target effects. In this case, by using a conditional Hhex knockout under a Fox A3 promoter, the fetuses showed a lack of hepatogenesis, as well as absent mesonephros and developmental retardation. During the two rounds of blastocyst complementation, 2 healthy fetuses were rescued on day 28 from 120 chimeric blastocysts. Immunohistochemistry confirmed that all hepatocytes were donor-derived, with enhanced GFP-labeled cells.

The above-mentioned examples are very recent examples of successful blastocyst complementation for liver development. While rates of chimerism and viable chimeras continue to pose a significant obstacle, advancements in this field, particularly using larger animals such as pigs, hold great promise for future interspecies blastocyst complementation.

## 3. Lung

Lung disease remains a significant cause of mortality and morbidity. The number of lung transplantations has progressively increased over recent decades, with over 3000 transplants being carried out in the US alone in 2023 [1]. The main causes for transplantation continue to be COPD (chronic obstructive pulmonary disease), cystic fibrosis and idiopathic pulmonary fibrosis, with the last one showing an increase in transplantation in North America [41]. Given the increasing demand for transplants, finding sources for lung tissue remains crucial.

### 3.1. Lung Development and Its Elimination

Lung development is a highly complex process with several steps resulting in significant cellular diversity. The initial development of the respiratory system starts from the primitive foregut endoderm. Initial elongation and bifurcation results in two primary bronchial buds that will eventually form the right and left lung. As the trachea develops and separates from the esophagus, mesenchymal cells surround and differentiate into cartilage precursors. The visceral and parietal pleura is formed from the splanchnic and somatic mesodermal layer, respectively. Functional lung tissue requires specific mesenchymal and epithelial cells and a complex interaction between the two as the epithelial components of an endodermal origin descend and undergo repetitive branching and growth into the neighboring mesenchyme [42]. Wnt/β-catenin (wingless-related integration site) has been shown to be crucial for this step, as inactivation has led to deviation in epithelial branching and differentiation between the distal and proximal lung, as well as decreased endothelial differentiation and mesenchymal growth [43,44]. Similarly, Nkx2-1 (Nk2 homeobox 1, also known as thyroid transcription factor 1, TTF-1) has been shown to be crucial for the branching of the bronchial tree [45]. Further branching, vascularization and differentiation (including eventual alveolarization) occur in the following weeks, allowing the progressive formation of future airways [42].

Several further key players have been identified in lung development. Fgf10 (fibroblast growth factor) and its major receptor Fgfr2b (fibroblast growth factor receptor) are one example. Specifically, Fgf10 has been shown to be essential for lung development as well as fore- and hindlimb formation. Indeed, even as early as E11.5, Fgf10-deficient mice showed no primary lung buds [46]. Moreover, Fgf10 has been shown to be highly involved in lung branching, with high, localized levels expressed in early lung development stages, demonstrating that it is important not only in initial, but also in further lung development [47]. Another notable factor is NKx2-1. While NKx2-1 is essential for thyroid, forebrain and pituitary development, it has also been shown to be crucial in several steps of lung development [45]. Firstly, it plays an important role in distinguishing the trachea and lung from the neighboring esophagus, without which trachea–esophageal fistulas will occur [48]. Furthermore, it is highly involved in lung morphogenesis, particularly distal branching, as well as the differentiation of specific epithelial cells in the lung [45,49,50].

Finally, additional important factors include Sox17 (SRY-box transcription factor), BMP (bone morphogenetic protein) and Foxa2. BMP has been shown to regulate proximal–distal differentiation [51], while Sox17 influences pulmonary vascular morphogenesis and the differentiation of respiratory epithelial cells [52,53]. Finally, Foxa2 has been shown to be pivotal for alveolarization and respiratory goblet cell expansion [54].

### 3.2. Application of Blastocyst Complementation

While the use of blastocyst complementation to generate lungs has remained a challenge for several years due to a lack of a definite gene target to prevent lung development, the past couple of years have shown great advancements in the area.

Mori et al. used conditional blastocyst complementation in two different mouse models [11]. The first targeted Fgfr2 gene deletion, resulting in the absence of lungs. Importantly, Fgfr2 is activated by several ligands, including Fgf10, and mice deficient in both have been shown to have a similar lung phenotype and limb defects [46,55]. The second mouse model investigated Wnt/β catenin (Ctnnb) knockout. The knockout of this pathway has been shown to lead to lung and trachea agenesis, compared to just lung agenesis with Fgfr2 [44,46]. They used donor pluripotent stem cells engineered to express a fluorescent protein (GFP) marking. In both models, the chimeric mice (4 Fgfr2-null mice, 3 Ctnnb1-null mice) showed normal lung development, with pulmonary function tests demonstrating comparable results to their wild-type counterparts. As expected, the GFP reporter demonstrated strong signals in the trachea and lung for the Ctnnb1-null model, while the Fgfr2-null model only showed strong signals in the lung. Interestingly, in Fgfr2-null mice, while AT1 and AT2 (alveolar type 1 and 2, respectively), secretory and multiciliate cells showed significantly high GFP signals, GFP signals in endothelial or mesenchymal cells were much more variable, with over half being of a host origin.

Further work has also been carried out by Kitahara et al. using Fgf-10-deficient mice, more specifically, Fgf10 Ex1^mut^/Ex3^mut^ heterozygous mutant mice [12]. As previously outlined, Fgf-10 is crucial for lung development. Once again, the ESCs that were injected for blastocyst complementation were marked with GFP. Initial work on neonate mice was promising and further work on mice that reached adulthood was completed. A total of 153 neonates were obtained from 638 microinjected embryos, with only 16 live chimeras surviving to weaning. All 5 Fgf10 Ex1^mut^/Ex3^mut^ chimeric mice survived and were eventually sacrificed at 4 months. No significant histological or morphological changes were observed in the adult Fgf-10 Ex1^mut^/Ex3^mut^ chimeras. Similarly, to Mori et al., Kitahara et al. showed strong GFP signals across the lung, especially in parenchymal cells. However, they also demonstrated strong reporter expression in interstitial regions, as well as vascular endothelial and smooth muscle cells. Overall, while a substantial number of cells were GFP-positive and, therefore, from donor cells, most cell types in the chimeric lungs showed a mixture of donor and host cells.

Miura et al. further highlighted the potential of blastocyst complementation using the Foxa2-driven Fgfr2 pathway as a knockout target [13]. Their aim was to find a single lineage that would encompass the lung epithelium and mesenchyme. They showed that using this pathway, instead of systemic Fgfr2 depletion, avoided agenesis in several other organs such as kidneys, limbs and more. Their initial work demonstrated that Foxa2 seemed to be involved in the majority of the lung epithelium and half of lung mesenchymal development. The lungs generated in the Foxa2-driven Fgfr2 knockout mice showed the rescue of the lung phenotype, with almost the entirety of the lung epithelium and mesenchyme being composed of the GFP-labeled iPSCs. Interestingly, the knock-out of Fgfr2 in the Foxa2 lineage led to decreased proliferation of host cells compared to donor cells in the mesenchyme, leading to increasing lung complementation during development. Follow-up was carried out 4 weeks post-birth and the proportion of host-derived cells was significantly low in both the epithelium, endothelium and mesenchyme. Pulmonary function testing showed no significant difference in the tidal volume, airway resistance, frequency and expiratory flow at a 50% expired tidal volume.

A different approach to blastocyst complementation has been examined, particularly via the Nkx2-1 pathway. Wen et al. first demonstrated promising results in 2020 in rescuing lung and thyroid tissue in Nkx2-1 homozygous knockout mice [14]. GFP-labeled mouse ESCs were injected into blastocysts. A histological analysis showed that rescued chimeric lung lobes contained bronchioles, blood vessels and alveolar saccules. Electron microscopy also showed the presence of AT2 cells with surfactant secretion and AT1 cells expressing T1α, both of which indicate successful differentiation during development. Furthermore, all respiratory epithelial cell subtypes showed similar gene expression, whether they were derived from ESCs or from endogenous cells. Importantly, most epithelial cells expressed GFP, while other cell types (endothelial, hematopoietic, fibroblasts and pericytes) showed more variable GFP signals. This demonstrated that a high ESC contribution remains a challenge in non-epithelial respiratory cells. Unfortunately, while lung and thyroid tissue could be rescued, tracheo-esophageal fusion was still present and complementation was inefficient in the forebrain. Furthermore, all Nkx2-1 homozygous knockout mice died at birth.

In 2024, Wen et al. showed further progress by using interspecies blastocyst complementation [15]. As expected, they showed that Nkx2-1 homozygous knockout rats lacked a lung and thyroid, with tracheo-esophageal fusion. Subsequent interspecies blastocyst complementation was performed using mice wild-type ESCs and analysis was carried out on chimeras aged E20.5. Similar to their previous study, the Nkx2-1 knockout mouse–rat chimeras showed that the mouse ESC-derived cells expressed AT1 and AT2 markers (T1α and Pro-SPC (prosurfactant protein C) and SPB (surfactant protein B), respectively). Furthermore, and in contrast to their previous work, in 30% of the Nkx2-1 chimeras, the mouse ESCs contributed highly to mesenchymal and vascular cells (including endothelial cells, smooth muscle cells, immune cells, pericytes and fibroblasts), with 98.5% of all cells in the new lung being mouse-derived. As before, however, the blastocyst complementation did not rescue the tracheo-esophageal phenotype. A high mouse contribution was found in both the forebrain and the thyroid. Finally, it was noted that the mouse–rat chimeras had both smaller body and lung sizes.

## 4. Kidney

Since 2000, the number of kidney transplants has progressively increased, with over 26,000 being carried out in the U.S. alone. Despite the increased number of procedures, the number of new people on the kidney transplant waiting list has increased to over 44,000 in the U.S. Of those on the waiting list, approximately 12% have been waiting 5 years or longer. The most common primary causes for kidney failure in adults on the waiting list have continued to be diabetes and hypertension [56]. While dialysis continues to be an important alternative for many patients, the increased quality of life and increased freedom in patients’ lives cannot be ignored. As such, blastocyst complementation offers up a unique opportunity to meet the continued demand for transplants.

### 4.1. Nephrogenesis Elimination of Kidney Development

Kidney development undergoes several stages, with the formation of the pro-nephros starting as early as the third week, with the subsequent development of the mesonephros, then metanephros. From there, the ureteric bud and metanephric mesenchyme develop, with the former eventually undergoing branching to form a basic renal structure. Both essential structures are of a mesodermal origin. The interaction between the ureteric bud and the metanephric mesenchyme is crucial for nephron formation. The Wnt pathway is particularly important for this interaction and allows for a mesenchymal–epithelial transformation and the differentiation of nephron epithelia [57,58].

Other major signaling pathways for nephrogenesis and ureteric bud branching include Sonic hedgehog, bone morphogenic proteins and fibroblast growth factors [58]. Further complex interactions between factors are required for nephrogenesis, including Lim1, Pax2 (paired box gene), Eya1 (eyes absent homolog), Six1,2,4 (sine oculis homeobox homolog), Sall1 (spalt like transcription factor) and WT-1 (Wilm’s tumor). Pax2 is essential for intermediate mesoderm differentiation and Pax2 knockout, for example, leads to the agenesis of the kidneys, ureters and genital tracts [59]. Both Eya1 knockout and Sal1 knockout lead to defects in the ureteric bud growth and, therefore, renal agenesis [60,61]. Wt1 knockout causes metanephric mesenchyme apoptosis, and hence no further kidney development [62]. Six1 knockout leads to metanephric induction failure and has been shown to interact with both Pax2 and Eya1, with the former likely acting downstream and the latter acting upstream to Six1. Six1 has also been shown to be required for both Pax2 and Sall1 expression [63].

### 4.2. Application of Blastocyst Complementation

Espejel et al. showed early promising results when injecting an FAH-deficient murine blastocyst with donor murine iPSCs. Their initial work investigated this technique for hepatocyte repopulation in FAH-deficient mice. Notably, an FAH deficiency also leads to severe kidney dysfunction due to oxidative injury in renal proximal tubules [64]. However, by injecting iPSCs in FAH-deficient mice, a significant repopulation of proximal tubular cells could be achieved, with over 50% being donor-derived. The chimeric mice also demonstrated a restored kidney function, with the creatinine levels showing no significant difference compared to wild-type mice [7].

Usui et al. used Sall1^−/−^ knockout mice in 2012 with mouse embryonic or induced pluripotent stem cells injected into the blastocyst cavity [16]. Interestingly, while the newly formed kidneys were entirely derived from the injected stem cells, the contribution was minimal in the bladder and ureter. Of the 9 neonate pups obtained, 3 were Sall1^−/−^. Of note, in the ESC-complemented mice, the nephron epithelia and renal stroma were almost fully made up of ESC-derived cells, while the collecting tubule showed a mix of cells. In the iPSC-complemented mice, of the 37 neonate pups retrieved, 5 were Sall1^−/−^. iPSC-derived cells largely contributed to all kidney epithelial cells, except for the collecting ducts. Furthermore, kidney stromal elements including vessels and nerves showed a mix of their cell origin. The contributions in other organ systems varied for both ESC- and iPSC-complemented mice; however, most non-kidney tissue did show some level of chimerism. In terms of development, kidneys were histologically and morphologically normal in ESC- or iPSC-complemented mice. Unfortunately, even complemented Sall1 knockout mice were not able to survive to adulthood, with many demonstrating similar issues with nursing as non-complemented Sall1 knockout mice. Moreover, they were unsuccessful in generating kidneys when injecting rat iPSCs into Sall1^−/−^ mice.

While the above-mentioned study did not find success with interspecies kidney generation, using the opposite configuration, i.e., injecting GFP-labeled mouse iPSCs and ESCs in rats, seemed to show promising results. Goto et al. demonstrated this, similarly, using Sall1 as a knockout target [17]. They firstly concluded that the failure in successful rat complementation in Sall1^−/−^ mice was due to insufficient and decreased efficiency in rat iPSC contribution to the metanephric mesenchyme. In the Sall1^−/−^ chimeric mice, the kidneys showed uniform GFP expression. Notably, the kidney size in mouse-complemented Sall1^−/−^ rats was smaller than wild-type or heterozygous rats, although they were similar in size to wild-type mice. They also demonstrated similar issues with mouse-derived contributions. While the metanephric mesenchymal cells (e.g., glomerular basal membrane, proximal tubule, loop of Henle) were entirely of a mouse origin, and in the correct localization, the collecting tubules and blood vessels contained a mix of cells. Importantly, a successful connection between the ureter and bladder was shown when injecting intra-urethral dye. Furthermore, they encountered similar difficulty with postnatal survival and all complemented Sall1^−/−^ rats had the previously mentioned nursing defect. This may be due to defects in olfactory development, potentially leading to an anosmic phenotype.

Finally, both Matsunari et al. and Wang et al. recently demonstrated promising results using human cells in pig embryos. Given their size, pigs represent an attractive option for growing human organs. Matsunari et al. used a Sall1 knockout model. While their first attempt using cloned embryos was unsuccessful, using IVF-derived embryos led to 1 chimeric fetus of 12 with morphologically and histologically normal kidneys, with positive donor-cell expression [8].

Given the difficulties in synchronizing development, Wang et al. opted to target both Six1 and Sall-1 as knockouts to ensure that a large window in developmental progression was achieved [18]. Following an injection of DsRed-labeled human iPSCs, gestation was terminated at either E25 or E28. This was, in part, due to concerns about potential contributions to brain tissue, as this approximately corresponds to E40 in humans, and neuron production has been shown to begin on E42 [65]. While no fully developed kidneys were obtained, when compared to wild-type embryos, the mesonephros obtained in the chimeric embryos showed a similar mesonephric density and were histologically similar. Furthermore, over 50% of the mesonephric cells showed DsRed expression, with higher rates of DsRed expression in mesonephric tubules and lower rates in mesenchymal cells. Importantly, DsRed-labeled mesonephric tubules expressed Sall1, Six1, Pax2 and WT1, all important kidney developmental markers, suggesting the potential for further successful development.

## 5. Pancreas

In 2021, there were almost 4000 people on the pancreas transplant waiting list in the U.S., with just under 1000 pancreases being transplanted that year. Most patients awaiting a pancreas transplant were patients with type 1 diabetes, with type 2 diabetes being the second most common cause. While oral medications and insulin delivery systems continue to be a mainstay for both these patient populations, the discrepancy between people on the waitlist compared to the number of transplants performed highlights the importance of pancreas transplants in patients with severe complications or unstable glycemic control [66].

### 5.1. Pancreas Development and Its Elimination

The pancreas develops from a ventral and a dorsal bud [67]. The dorsal bud is of an endodermal foregut origin and will form the tail, body and neck of the pancreas, as well as the duct of Santorini. The ventral bud arises from the hepatic diverticulum and forms the head, the uncinate process and the duct of Wirsung. Importantly, there are initially two ventral buds; however, the left one must regress to prevent the formation of an annular pancreas [68,69]. The fusion of the remaining ventral bud with the dorsal bud occurs around week 6–7 during gut rotation to form the whole pancreas. Endocrine cells are initially identified shortly after this process and islet formation slowly progresses with insulin-expressing cells developing first. The proportion of insulin, glucagon and somatostatin, γ- and ε-producing cells resembles that of adult pancreases by week 21. That is, about 60% of cells are insulin-producing, 30% are glucagon-producing, and the other 10% are made up of the remaining cell types [67,69,70].

Signaling for pancreas development involves several pathways. Studies have detected Pdx11 (pancreatic-duodenal homeobox) and SHH (Sonic hedgehog) around week 4. Sox9, GATA4 and 6 have also been identified around week 5. The expression of certain factors becomes much more important in differentiating cell types, with NEUROG3 (Neurogenin) being a hallmark of endocrine cells, GATA4 of acinar exocrine cells and Sox9, Pdx1 and FoxA2 being found in pancreatic duct cells. While NEUROG3 seems to be required to determine an endocrine cell fate, it is only transiently increased and no longer detected by 35 weeks [67,71]. The inactivation of Pdx1 causes pancreatic agenesis and an exocrine deficiency [72]. Similarly, changes in downstream regions of PTF1A (pancreas transcription factor) have also led to pancreatic agenesis [73]. Further potential targets that may cause defects in pancreatic development include GATA4 and 6, HNF1B, Sox9 and UBR1 (ubiquitin protein ligase E3 component n-recognin) [71].

### 5.2. Application of Blastocyst Complementation

Current diabetes treatment, particularly for type 1 diabetics, requires lifelong glycemic control and insulin injections. As such, pancreas and pancreatic islet transplantation have become an important aspect of treatment for patients with difficult glycemic control, as well as patients with severe complications. For type 2 diabetics, while islet transplantation is not currently available due to their increased need for insulin, pancreas transplantation remains an option for a subset of these patients, with reported successes [74]. As such, there is a continued need for pancreatic organs or tissue, which cannot be met with the current supply [66].

Some success has been shown using the xenotransplantation of islets. This was first carried out in 1994, where 10 type 1 diabetic patients with a kidney transplant (and, therefore, immune suppression) received porcine pancreatic tissue. Porcine C-peptide was detected for 200–400 days in the urine of four patients [75]. Interestingly, another case study demonstrated that in a patient with type 1 diabetes who underwent porcine islet transplantation, live cells were found and retrieved 9.5 years following the transplant, with insulin production being confirmed in in vitro glucose stimulation [76]. Therefore, while xenotransplantation shows potential, the use of blastocyst complementation provides a unique opportunity, as it allows for a decreased need for immune suppression and may increase overall functionality.

Kobayashi et al. were some of the first to demonstrate success with blastocyst complementation back in 2010 [19]. They first demonstrated that blastocyst complementation could be used for organ generation. Using a Pdx1^−/−^ blastocyst, they developed a niche for pancreas development. Mouse iPSCs or ESCs were then injected, which led to morphologically and histologically normal pancreas generation, with pancreatic islets, duct epithelia and exocrine tissue being entirely derived from the mouse donor cells. However, as with other organs, stromal elements such as vessels, fibrocytes or nerves showed a mix of host and donor cells. Functionally, glucose tolerance testing showed successful insulin secretion in response to glucose with the maintenance of normal glucose levels. They further highlighted the potential of blastocyst complementation by transplanting the iPSC-derived islets into mice with induced diabetes. No immune suppression was required and the transplanted mice showed normal blood glucose levels and a normal response to glucose tolerance testing. Most importantly, they went on to produce interspecies pancreases. Rat iPSCs were injected into mouse blastocysts. The pancreatic epithelia of all 10 Pdx1^−/−^ mice was entirely made up of the rat-derived cells. While the number of chimeric mice reaching adulthood was low, in the 2 that did fully mature, their pancreases were histologically and morphologically normal with normal serum glucose levels and response to glucose loading. Their work was critical in highlighting that blastocyst complementation could be successful for the pancreas, even in the case of inter-species complementation.

The same group went on to demonstrate that the opposite was possible, that is, using EGFP-labeled mouse iPSCs or ESCs to rescue the pancreatic phenotype in Pdx1^−/−^ rats. The pancreatic endocrine, exocrine and duct epithelium showed EGFP expression. The Pdx1 knockout chimeric rats responded more slowly in the glucose tolerance test but otherwise showed no significant functional changes. They then transplanted the mouse-derived pancreases into diabetic mice. As has been previously discussed in other organs, they did find that a notable amount of tissue, such as endothelial cells, were rat-derived. Therefore, while the transplanted mice were given immune suppression for the first five days, the transplanted mice showed normal glycemic levels for over 370 days even without continued immune suppression. Transplanted islets expressed insulin, glucagon and somatostatin, underscoring the potential for blastocyst complementation as a treatment modality [21].

Matsunari et al. further demonstrated success in the area; however, they used a larger pig model [20]. Firstly, they demonstrated that Hes1 (hairy and enhancer of split) overexpression under Pdx1 promotion led to the inhibition of pancreatic development. They then showed that blastocyst complementation could rescue pancreas development in these pigs. Of the 14 full-term fetuses obtained, 5 were chimeric, with histologically normal pancreases and almost all pancreatic cells derived from donor cells. They went on to successfully generate chimeric adult pigs, with normal serum glucose levels. The oral glucose tolerance test results were also normal in one of the chimeric pigs. Necropsy of one chimeric pig showed macroscopic normal intestinal organs. While all tissue examined in this study showed donor-derived cells, the progeny sired by the chimeric male pigs all demonstrated an apancreatogenic phenotype, suggesting that the sperm were derived from host, not donor cells.

Finally, more recently, Matsunari et al. further continued their work in another pig model [8]. However, whereas their work in 2013 was carried out using Pdx1-Hes1 overexpression, in this case, they used a PDX1 knockout model. Of the 10 fetuses obtained, 4 were chimeric, with half of the chimeras showing a histologically and morphologically normal pancreatic phenotype. Compared to the chimeras with non-rescued pancreases, these pigs showed high levels of chimerism in the pancreas.

The above-mentioned studies have not only shown that blastocyst complementation could be successful for pancreas generation, be it in small or large animal models, but that the interspecies transplantation of generated organs could also improve glycemic control in diabetic animals.

## 6. Heart

Heart transplantation is a life-saving operation for many people. There continues to be an unmet need for transplant donors, with over 8,000 people in the USA on the waiting list in 2022 alone. The leading cause for requiring a heart transplant remains cardiomyopathy, with coronary heart disease coming in second. Approximately 400 transplants have been done in the pediatric population and congenital defects continue to be the leading cause of heart transplants in this patient population [77]. The need for donor hearts, therefore, remains critical for many of these patients and, as such, finding possible alternatives continues to be of great importance.

### 6.1. Cardiogenesis and Elimination of Cardiac Development

Cardiac development starts from mesodermal cells, with growth factor secretion (such as BMP) from neighboring endodermal and ectodermal cells allowing for differentiation to occur [78]. Concomitant to cardiomyocyte development, the endocardium develops from a specific subset of mesodermal progenitors. This process seems to be largely dependent on Tie and Tie2 expression, with the knockout of these genes leading to normal vascular structures except for the endocardium [79,80]. Further development leads to folding, with the heart tube organizing into a 2–3 cardiomyocyte layer and one inner endocardial layer. Subsequent looping allows for the formation and spatial relation of a four-chambered heart, with cardiomyocyte differentiation into either atrial, ventricular or conduction-specific cells [81].

Several transcription factors have been identified to be critical for cardiac development from the mesoderm, including Nkx2-5, GATA4 and MesP1 (mesoderm posterior bHLH transcription factor). With regards to Nkx2-5, while its deletion did not lead to agenesis, looping was disrupted, as well as trabeculation and endocardial cushion formation [82]. Further differentiation also seems to be greatly affected by GATA transcription factors, with GATA4 inhibition causing defects in cardiomyocyte differentiation and further development [83,84]. Finally, MesP1 and P2 have been shown to be important for mesodermal migration from the primitive streak, with a MesP1 deficiency leading to the inhibition of cardiac mesoderm [85,86].

### 6.2. Application of Blastocyst Complementation

The first hurdle in applying blastocyst complementation for cardiogenesis is that no single knockout target has been identified that fully inhibits cardiogenesis. While Coppiello et al. showed success in the area, this required the depletion of both the heart and vascular system using Cre (cyclization recombination enzyme)-dependent DTA (diphtheria toxin subunit A) to mediate agenesis [22]. In order to induce cardiomyocyte agenesis, they used a Nkx2.5-Cre strain, and to mediate endothelial agenesis, they used a Tie2-Cre strain. Firstly, they demonstrated that cardiac complementation could be achieved when injecting mouse iPSCs into Nkx2.5-Cre mice. The hearts in all 14 live embryos obtained on embryonic day 14 showed normal beating and morphology, with 100% of the cardiomyocytes being donor-derived. Per the vascular complementation in the Tie2-Cre strain, of the 42 chimeras produced, 5 were successfully Tie-Cre (tyrosine kinase with immunoglobulin-like and EGF-like domains)-complemented with a normal morphology and practically all endothelial cells were of donor origin (99.3%). Combining both Tie2-Cre and Nkx2.5-Cre strain complementation led to the successful development of chimeric mice with a donor-derived heart and vascular system. The heart function (via echocardiogram) as well as the vascular and cardiomyocyte density in 8 and 3 chimeras, respectively, showed no difference when compared to the control animals. The interspecies complementation of rat ESCs in mouse blastocyst proved to be difficult. While they achieved rat heart complementation in 19/47 Nkx2.5-Cre-strain mice at E10.5, with practically all cardiomyocytes being donor-derived, they were unable to show any success with heart or vascular system complementation in Tie2-Cre or Nkx2.5-Cre mice at E11.5 or E14.5. Difficulty in this area seems to be in part due to improper vascularization and vascular development to allow for appropriate oxygenation.

## 7. Thyroid

Hypothyroidism continues to be one of the most common endocrine diseases. More recent data analysis suggests that just over 10% of the U.S. population suffers from this disorder [87]. The mainstay of current treatment for hypothyroidism continues to be thyroid hormone replacement therapy. While inexpensive, and relatively accessible, this is not without its own challenges, as treatment is lifelong and requires continued thyroid testing and monitoring. Particularly in older patients, over or under treatment with thyroid replacement therapy can have significant effects. In one study, only 42.8% of patients under thyroid hormone treatment had TSH in the euthyroid range, with 41% having low TSH and 16% having high TSH [88]. This can have clinical significance, particularly in older patients, where it has been shown that patients under replacement therapy with TSH outside of the euthyroid range can lead to adverse health effects [89,90]. Thyroid transplantation, or thyroid tissue transplantation, has become an area of interest, especially for patients where thyroid hormone replacement therapy can prove challenging [91]. As such, the use of blastocyst complementation also offers up an interesting treatment alternative for this patient population.

### 7.1. Thyroid Embryogenesis and Elimination of Thyroid Development

The thyroid gland is of an endodermal origin from the ventral pharynx wall near the base of the tongue. The primitive thyroid diverticulum eventually migrates down to take its spot right above the trachea, as well as bifurcating to form 2 lobes [92,93]. Thyrocytes, which are of an endodermal origin, start forming follicles as the left and right lobe are established. The calcitonin-producing C cells, which were initially thought to be of a neural crest origin, may in fact also be endoderm-derived [94]. Transcription factors that have been found to be crucial for thyroid development include Nkx2-1, Pax8, Fox1 and Hhex [95]. Nkx2-1 and Pax8 have been shown to be enough to determine the differentiation of thyroid follicular cells, although a deficiency in either one of the four above-mentioned transcription factors will lead to severely impaired morphogenesis or agenesis [96,97].

### 7.2. Application of Blastocyst Complementation

Ran et al. already demonstrated that blastocyst complementation to generate lungs was possible using Fgf10 Ex1^mut^/Ex3^mut^ mice [12]. They once again used this model, which led to the development of severely hypoplastic thyroids, with only remnant tissue being present. Of note, targeting the Fgf10 pathway did not seem to influence calcitonin expression. GFP-labeled mouse iPSCs were then injected into the embryos. An initial analysis of neonatal chimeras showed morphologically and histologically normal thyroids with high GFP expression and comparable Tg (thyroglobulin), T3 (triiodothyronine) and Nkx2-1 expression when compared to wild-type mice. Five complemented chimeric mice that survived to adulthood were then further analyzed. The expression levels of Nkx2-1, Fox1 and Pax8 in the newly generated thyroid follicular cells were similar to levels found in wild-type mice. The histology was also normal. The proportion of donor derived cells amongst thyroid follicular cells was greater than 85%. Notably, mesenchymal and C-cells did not show a similar donor-derived predominance. Finally, functional testing demonstrated comparable levels of T3 and T4 (thyroxine) between the wild-type and chimeric mice [23].

Wen et al. more recently also demonstrated that blastocyst complementation could rescue the thyroid phenotype in Nkx2-1 knockout mice [14]. They initially investigated this pathway to determine whether blastocyst complementation could rescue mice with lung agenesis; however, since this pathway also leads to thyroid agenesis, both organ systems were evaluated together. Nkx2-1^−/−^ mice were injected with GFP-labeled mouse ESCs. At embryonic day E17.5, the chimeras showed thyroid tissue, with the majority of thyrocytes expressing donor-derived GFP. Unfortunately, no functional testing could be performed in adult mice, as tracheo-esophageal separation could not be achieved in the chimeric mice, and all died at birth.

## 8. Others: Parathyroid, Thymus

Finally, there are a variety of further solid organs that have also been investigated. The application of blastocyst complementation for these has, however, been more limited compared to the liver, lung, kidney, heart and pancreas. The organs discussed below include the parathyroid and the thymus. The successful use of blastocyst complementation in these instances continue to demonstrate the potential that this technique can have in a multitude of systems.

### 8.1. Development of Thymus and Parathyroid

The thymus and parathyroid both originate from the endodermal gut tube, although the ectoderm also contributes to thymic epithelium. The thymus and one pair of parathyroid glands will eventually arise from the third pharyngeal pouch, with a specification of the cell fate leading to parathyroid cells assuming GCM2 positivity (glial cells missing transcription factor) and thymus cells assuming FoxN1 positivity. The second pair of parathyroids originate from the fourth pharyngeal pouch [98]. The two thymic lobes eventually gravitate toward the midline in order to fuse together and then reach their final position in the anterior superior mediastinum. Hematopoietic progenitor cells eventually migrate to colonize the thymus, with stem cells from the bone marrow and fetal liver, at which point the thymus assumes its critical role in T cell maturation and development [99,100].

The parathyroids from the third pharyngeal pouch similarly undergo caudal migration to eventually become the inferior parathyroids, while the parathyroids from the fourth pharyngeal pouch only minimally move and will form the superior parathyroids. The chief cells of the parathyroid will ultimately start producing PTH (parathyroid hormone) [98,100].

GCM2 is crucial for parathyroid cell differentiation as well as preventing apoptosis. GCM2 activation is also largely dependent on SHH signaling from the endoderm or neighboring mesenchyme, which likely acts via GATA3 and Tbx1 (T-box) downstream activation. Other transcription factors that seem to play a critical role in parathyroid organogenesis include HOX3 (homeobox), which may work together with PBX1 (pre-B-cell leukemia transcription factor), Pax, Eya and Six transcriptional regulators to enable parathyroid development. Of note, MafB (musculoaponeurotic fibrosarcoma oncogene B) in particular seems to be required for the eventual production and expression of PTH [98].

The thymus requires many of the same transcription factors mentioned above for successful development. The earliest marker for specific thymic differentiation is Fox N1 and it has been shown to be important for branching and thymic colonization, although it does not seem that it is required for initial thymus formation or migration. Early thymus development seems to involve HOXa3, Eya1, Pax1/9, Six1/4 and Tbx1 networks, with the clear involvement of Wnt and BMP signaling, although there is likely a still unclear driving force for the very initial thymus fate development [99,100].

### 8.2. Application of Blastocyst Complementation

Given the importance of the Fox N1 pathway in thymic epithelial cell differentiation, Yamazaki et al. used a Fox N1 knockout model to generate athymic mice [24]. The knockout mice were complemented with mouse ESCs. The extent of chimerism did not seem to influence thymic rescue, although lower chimerism seemed to result in smaller thymus. Of note, almost all thymic epithelial cells were donor-derived, with a higher percentage of peripheral T cells being donor-derived if chimerism levels were higher. When compared to normal mice, there was no significant difference in the number of peripheral T cells or in the gene-expression profile of complemented thymic epithelial cells. Of note, neither donor nor host-derived thymic cells showed significant differences in the proliferation of CD4+ and CD8+ T cells upon stimulation with antibodies, and chimeric mice showed a similar production of IFNγ (interferon), IL-2 (interleukin) and Granzyme B by splenic T cells. Finally, anti-PDL1 antibody (programmed death-ligand) treatment in the chimeric mice increased T cell activation and IFNγ production, as well as suppressing tumor growth. The results suggested that blastocyst complementation led to the rescue of a functional thymus.

Miura et al., while investigating Foxa2-driven Fgfr2 depletion for lung agenesis, also found that blastocyst complementation successfully rescued the thymus agenesis phenotype [13]. In 5 of the Foxa2-driven Fgf2r knockout mice, an average of 92.4% of the thymic epithelium and 52.9% of the thymic mesenchyme was chimeric (although the latter showed much greater variability in the extent of chimerism).

Kano et al. very recently demonstrated functional parathyroid gland generation using blastocyst complementation with a GCM2 knockout model and complementation with mouse ESCs [25]. Histologically normal parathyroid glands were achieved, with donor-derived chief cells. Other cell lineages, such as endothelial or mesenchymal cells, showed a mix of donor and host-derived cells. Functionally, compared to control mice, complemented mice showed similar plasma calcium levels, basal PTH values and PTH stimulation response. The gene expression profiles showed either higher or comparable expression levels. The transplantation of the donor-derived parathyroid glands into GCM2 post-natal knockout mice led to survival in the transplanted group. Finally, they achieved interspecies blastocyst complementation using rat hosts and mouse ESCs. The GMC2 knockout rats were injected with mouse ESCs, leading to the development of parathyroid glands which exhibited transcription factors required for further development and PTH expression. Unfortunately, due to the development of umbilical hernias, survival after birth was not possible.

## 9. Challenges of Interspecies Chimerism

Blastocyst complementation is a promising option for organ generation, especially for transplant use. This review underlines the many encouraging results for several solid organs. However, there are certainly challenges that must be overcome in this field.

### 9.1. Inefficient Chimerism and Survival to Adulthood

In all studies discussed, there were difficulties in generating living chimeras that survived to adulthood. Such survival can be very challenging when knockouts of major pathways lead to phenotypes that cannot be rescued with blastocyst complementation and simply do not allow for postnatal survival (e.g., tracheoesophageal fistula in Nkx2-1 knockouts, nursing issues in Sall-1 knockouts or umbilical hernias in GMC2 knockouts) [14,15,16,17,25]. Furthermore, success with interspecies chimerism seems to be inversely proportional to the evolutionary distance between the donor and host species [101]. The most successful interspecies chimerism to date has been observed between evolutionarily closely related species such as the mouse and rat, which have been outlined in this review [15,17,19,21,22,25].

Inefficient chimerism was particularly apparent with respect to donor contributions in endothelial and vascular systems. For most of the solid organs, host tissue still largely made up these anatomical areas. This has great significance for interspecies chimerism, particularly when considering the risk of rejection with transplantation of the chimeric organ into the donor recipient. Being able to combine vasculature depletion with other organ-of-choice depletions would allow for overall higher levels of chimerism in the newly generated organ, particularly in supporting tissue. Matsunari et al., for example, demonstrated that the blastocyst complementation of dual knockout KDR (kinase insert domain receptor to inhibit vasculogenesis) and Pdx1 (to inhibit pancreatic development) led to a fully developed pancreas with donor-derived endothelial and hematopoietic cells in four full-term pigs [8].

As such, the successful development of donor endothelial and vascular cells using blastocyst complementation is a field of great interest, even for solid organ generation. Hamanaka et al. successfully generated almost 100% donor-derived vascular endothelial and hematopoietic cells in Flk-1 knockout mice (also known as VEGFR2, vascular endothelial growth factor receptor). Furthermore, the successful chimeras were able to survive to adulthood. Unfortunately, interspecies blastocyst complementation using rat iPSCs in host mice were unable to survive past E13.5 due to incomplete complementation [102]. Recently, Das et al. successfully generated entirely human-derived hemato-endothelial cells in ETV2-null pigs analyzed between embryonic day 17 and 18, an important step for the inter-species generation of hemato-endothelial cells [103]. Successes in developing hemato-endothelial cells are of great significance and will prove to be important with regards to developing organs with the greatest number of donor-derived cells as possible, an invaluable step to avoid graft rejection when organs are transplanted back into the donor.

Successful interspecies chimerism becomes even more evident with more evolutionarily distant donor–host species’ pairs, such as human–mouse and human–pig. To date, these combinations have exhibited low levels of donor contributions to the host and low success rates of chimerism. As an example, with human donor cells in a MYOD/MYF6-KO (myoblast determination protein/myogenic factor) pig model, <5% of skeletal muscle was derived from human cells [104]. Certain measures have been shown to improve human contribution in interspecies models, such as the inhibition of apoptosis. BMI1 (B-lymphoma Mo-MLV insertion region) and BCL2 (B-cell lymphoma) overexpression to inhibit apoptosis allowed for the increased efficiency of human contributions in mouse, rabbit and pig models. Introducing human iPSCs or ESCs in pre-implantation blastocysts, even when stage-matched, has been shown to be difficult, likely as even in their “naïve” state, these cells are more primed than ESCs or iPSCs from other species, leading to increased apoptosis when injected. Therefore, overexpressing anti-apoptosis factors such as BMI1 and BCL2 can help improve this initial hurdle and result in improved survival and efficiency in interspecies chimera generation [105,106].

Finally, another aspect that must be considered is the type of donor stem cell that is used during blastocyst complementation. These donor cells must be effectively integrated in the inner cell mass of host blastocysts. For example, murine ESCs are derived from the inner cell mass in blastocyst-stage embryos, while iPSCs are reprogrammed somatic cells that have induced pluripotency by overexpressing pluripotency transcription factors [107]. As such, this may impact their differentiation potential. Such considerations also hold true when one considers interspecies differences in the stem cells used. As previously mentioned, human PSCs are not considered to be as “naïve” as their murine counterparts. In fact, human PSCs likely represent the equivalent of murine epiblast stem cells from a post-implantation, rather than a blastocyst stage, and are, therefore, in a more advanced primed pluripotency state and may explain why contributions to blastocyst chimeras with human PSCs can be more challenging [108,109,110]. The developmental stage of these stem cells, therefore, has a great effect on the efficacy of contribution in chimeras. For example, in mice, naïve pluripotent cells have shown greater success compared to primed epiblast stem cells. Similarly, primed human PSCs were inefficient when compared to their naïve-like PSCs. Particularly for human PSCs, a functional assessment may be difficult and largely relies on molecular markers and criteria [109,110]. While this has enabled some differentiation between naïve and more advanced or intermediate-stage cells, characterizing the type of stem cells that are used is of great importance for future success. Finally, there may be further interspecies differences at play. Aksoy et al. demonstrated that, as expected, mouse ESCs showed the efficient colonization of both rabbit and monkey embryos. In contrast, monkey and human naïve PSCs were much less effective in colonizing both host embryos, with monkey PSCs demonstrating arrested growth in the G1 phase and premature differentiation [111]. In short, it will be crucial to understand the differences in stem cells from distinct species, as well as the developmental stages that these cells represent.

### 9.2. Barriers to Interspecies Chimerism During Development

Some of the major barriers for successful interspecies chimerism include (i) cell competition between the donor and host cells leading to the limited contribution and cell death of donor cells in the chimeric embryo; (ii) issues with donor–host ligand and receptor compatibility; (iii) asynchrony in developmental speeds between the donor and host cells in the developing chimera; and (iv) a mismatch in developmental stages of donor and host cells at the time of the introduction of donor cells into the host embryo [112,113,114]. Multiple strategies have been employed to overcome these barriers, with infrequent success.

The limited contribution and cell death of donor cells in chimeric embryos has been discussed above, including the use of the overexpression of anti-apoptotic genes allowing for increased human contributions in chimeras. Other strategies like matching developmental stages of donor and host species, during the introduction of donor cells into the host embryo, have been explored. Stage-matching chimerism studies have demonstrated that successful chimerism was observed when (i) mouse ESCs, a pluripotent cell type, were injected into the early mouse blastocyst (E3.5); and (ii) mNCCs (mouse neural crest cells), a multipotent cell type, were injected into the late mouse E8.5 gastrulating embryos [115,116]. However, when the donor and host stages were interchanged and the early mESCs were injected into the late gastrulating embryo, or the multipotent mNCCs were injected into the early E3.5 embryo, successful chimerism was not observed. This mouse–mouse stage-matching study demonstrated the importance of matching developmental stages between donor cells and host embryos. The naïve mouse ESCs were more compatible with the early-stage mouse blastocyst and more differentiated donor cells contributed better to later-stage gastrulating embryos where the germ layers were defined.

Another study of interspecies chimerism demonstrated that human pluripotent stem cells (PSCs) injected into the E6.5-7 gastrulating mouse embryo successfully contributed to the in vitro cultured chimeric embryo and had cell-type-specific gene expression [117]. Since human PSCs are more ‘primed’ than the ‘naïve’ mESCs, the h-PSCs are at a later stage of development and are more like the primed cells from the mouse post-implantation embryos [118,119]. The authors observed the successful proliferation of human cells in the mouse embryos in 70% of the chimeras [117]. However, these embryos were cultured in vitro for only 2 days and, hence, any further inferences could not be made.

While these experimental strategies have been useful in demonstrating interspecies barriers and overcoming them to some extent, these trial-and-error ex vivo and in vivo procedures involving embryos are expensive, time consuming and low-throughput. Very few studies have interrogated molecular factors that are crucial for enhancing chimerism, identifying targets to ‘synchronize’ donor–host cells and overcoming barriers in the less successful chimerism pairs. There is a need to understand the cellular and molecular signaling mechanisms of the donor and host cells as they develop together within the chimeric embryos. Furthermore, there is a need to be able to do so in an affordable, high-throughput manner.

#### 9.2.1. Single-Cell Molecular Approaches to Understand Donor and Host Cell Mechanisms in Chimeric Embryos

##### Single-Cell RNA Sequencing for Enhancing Interspecies Chimerism Efficiency

The development of sc (single cell) sequencing technologies, such as sc-transcriptome and sc-genome sequencing, has revolutionized the way in which we are able to gain insight into the genome features. Sc genome sequencing techniques allow us to gain insight into genome coverage, copy number variation at the level of individual genes and single nucleotide variations. One of the major barriers to successful interspecies chimerism is the differences in developmental stages between the donor and host cells [114]. For example, the donor and host cells might not match with regards to the level of differentiation, embryonic stage or donor stem cell status (naïve versus primed). Using sc-RNA sequencing, we can analyze gene expression across early embryonic stages and stem cells and computationally stage-match donor and host species based on gene expression [120]. Sc-RNA sequencing has been performed on multiple cell types, including early developing embryos of humans, mice, pigs, marmoset and other species [121,122,123,124]. Access to these datasets allows us to perform the in silico stage-matching of multiple donor–host species [120]. In Shetty et al., 2023, the authors used an in silico analysis approach, based on similarities in gene expression, to stage-match stem cells and early embryos of commonly used donor and host species, including human, marmoset, mouse and pig [120]. They identified that the stages that best matched with each other were the human blastocyst (E6/E7), the gastrulating mouse embryo (E6–E6.75), the late marmoset inner cell mass and the late pig blastocyst. They also found that the human SCs best matched with the late mouse gastrulating embryo, in line with the results of the ex vivo and in vivo stage-matching experiments previously performed [117,125,126]. This is a high-throughput, fast and cost-efficient analysis that narrows down multiple donor cell–host embryo combinations that have the most similar gene expression and hence are most likely to produce successful chimeras when tested using ex vivo and in vivo techniques.

##### Replication Timing to Determine Developmental Speeds of Donor and Host Species

In addition to sc-transcriptomics, sc-genomic DNA sequencing allows us to analyze patterns of genome replication through a technique called ‘replication timing’ [127,128]. RT (replication timing) provides us with information on how the cell’s genome replicates during the S-phase of the cell cycle. This allows us to gain insight into different genes and if they replicate early or late during the S-phase. RT patterns are conserved within cell types, mitotically inherited and correlated with chromosomal organization and gene expression [129,130]. Using RT, one can also gain insight into not only how the cells replicate, but also how they coordinate with the cell-type-specific speed of replication.

One of the major barriers to successful chimerism is the differential developmental speeds of the donor and host cells in the chimeric embryo (Figure 2) [113,120]. Currently, there is no established way to define the ‘developmental speed’, other than the metric of hours and days. Using RT, it would be possible to define the cell-type-specific developmental speed at the molecular level and further analyze differences in the donor–host developmental speeds at the genomic level. This will enable us to understand the current developmental speed barriers in the interspecies chimerism field and allow us to identify molecular targets to overcome these barriers.

### 9.3. Ethical Considerations

Finally, a particularly important note in using human interspecies models must consider the human cell contribution in a developing embryo and, in particular, brain tissue. As such, timing must consider the development of off-target tissues and identify the best genes to target for minimal extra-organ contributions. In models where major pathways such as Nkx or Hhex are targets for knock-out, the levels of chimerism in other organs, particularly in neural or germ-line tissue, are particularly important to consider. Kobayashi et al., for example, induced Mixl1 (mix paired like homeobox) expression to attempt to restrict injected stem cell differentiation to endodermal tissue. They found that at a sufficient level, this did result in a significant reduction in the donor pluripotent stem cell contribution to organs that were not of an endodermal origin [131]. Hashimoto et al. further demonstrated that the knockout of Prdm14 (PRDI-BF1 and RIZ homology domain-containing) and Otx2 (orthodenticle homeobox protein) inhibited the contribution of donor cells to germ-line spermatozoa, oocytes, testes, ovaries as well as the brain [132]. These studies underscore that certain measures can be taken to address ethical concerns for human interspecies chimeras; however, addressing these early is of great importance. Furthermore, in human organs developed in animals, particularly if they are not 100% human-derived, there may be concern over zoonotic diseases following transplantation back into the human host. However, screening programs and a robust understanding of possible host pathogens could overcome such hurdles [133,134].

There are also several legal considerations. While current U.S. federal laws do not explicitly restrict human interspecies chimers, there is still a moratorium in place for the federal funding of human–animal chimera research [135]. The most recent report from the National Academies’ Human Embryonic Stem Cell Research Advisory Committee from 2010 still states that research whereby the breeding of animals where human ESCs’ or iPSCs’ introduction may contribute to the germ line continues to be ineligible [136]. Concrete ways of ensuring such a contribution does not occur during blastocyst complementation remains an ethical, but also a legal and regulatory necessity. Finally, should interspecies blastocyst complementation become a viable option for the generation of donor organs, institutional reviews and regulations will likely need to be put in place, particularly regarding the distribution and ownership of generated organs. This will likely prove even more challenging, as acquiring viable transplant organs will be a time-sensitive matter for patients, particularly when considering the time required for complete animal and organ development to adulthood. In short, prior to the use of chimeras for donor organ generation in the field of transplantation, several ethical and legal issues will need to be addressed.

## 10. Conclusions

While many challenges and questions remain in the field of blastocyst complementation, significant progress has been made. A number of solid organs have been successfully generated using this technique, with significant efforts dedicated to improving chimera generation and addressing ethical concerns. As a result, blastocyst complementation continues to present a promising avenue to meet the ongoing demand for organ tissue in the transplant and medical field.

## Figures and Tables

**Figure 1 genes-16-00215-f001:**
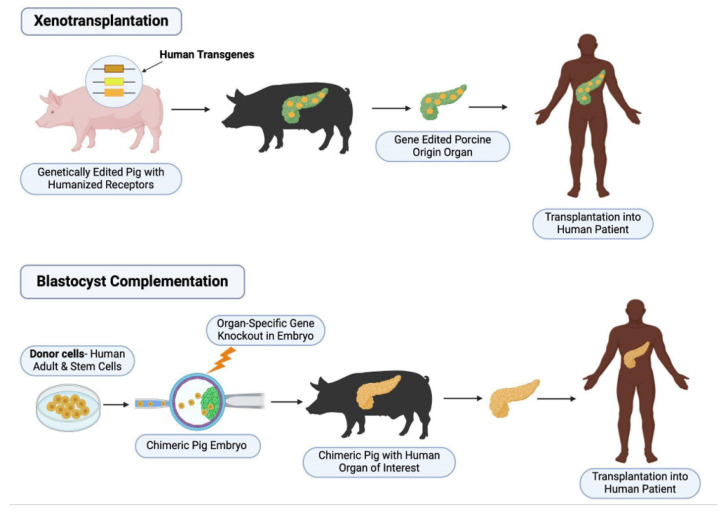
Two Distinct Genetic Approaches to Chimeric Organ Transplantation.

**Figure 2 genes-16-00215-f002:**
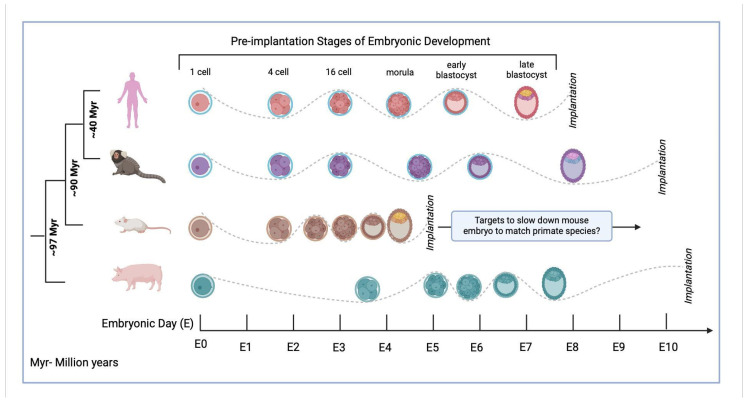
Barrier to Interspecies Chimerism—Differences in Developmental Speeds Across Donor and Host Species. Early preimplantation embryos of human, marmoset mouse and pig embryos are depicted from the zygote to the late blastocyst stage.

**Table 1 genes-16-00215-t001:** The summary of results from all 19 studies, including details on the organs generated, the host and donor species, the cells used for complementation, the target gene modifications (such as knock-out [KO] or gene introduction) to achieve organ agenesis, the functionality of the de novo organs and the survival or follow-up timelines in chimeric animals (if animals were studied prior to birth, time will be denoted with E, for embryonic day). Studies that have included interspecies blastocyst complementation are highlighted in **bold**.

Organ	Species	Gene Target	Function	Survival/Follow Up	Reference
Liver	Mouse host, mouse donor	FAH KO	6/40 chimeric mice had high enough chimerism for survival without requiring 2-(2-nitro-4-fluoromethylbenzoyl)-1,3-cyclohex-anedione for treatment of liver failure. Highly chimeric mice showed full hepatocyte repopulation by day 70, with 80% FAH expression in iPSC-derived hepatocytes when compared to wild-type mice. Liver function testing was normal compared to wild-type mice at day 70 and 300 (albumin, bilirubin, alkaline phosphatase, alanine aminotransferase).	Day 300	Espejel et al. [7]
Liver	Pig host, pig donor	Hhex KO	Of the 4 chimeric fetuses achieved in the first round of blastocyst complementation, 1 showed normal liver. In the second round, 3 additional chimeric fetuses from 95 complemented blastocysts were alive at cesarean.	Full-term development, alive at time of cesarean	Matsunari et al., 2020 [8]
Liver	Pig host, pig donor Mouse host, mouse donor	Hhex KO	In pigs: restoration of Hhex and AFP liver-protein expression in liver cells, with high donor eGFP+ signaling in liver tissue. In mice: Increased survival past the embryonically lethal stage in Hhex knockout, with retarded-to-normal growth when compared to age-matched wild-type embryos. High degrees of chimerism present in complemented embryos.	Pig: E25 Mice: E12.5	Ruiz-Estevez et al., 2021 [9]
Liver	Pig host, pig donor	Conditional Hhex KO with Fox A3 promoter	Two rounds with each 2/120 healthy fetuses with all hepatocytes of donor origin.	Day 21	Simpson et al., 2024 [10]
Lung	Mouse host, mouse donor	Ctnnb, Fgfr2 KO	Pulmonary function tests (resistance, compliance, elastance, methacholine challenge) showed non-significant differences between wild-type and Ctnnb-null and Fgfr2-null. GFP+ donor-derived signals were very strong in epithelial tissue, but variable in mesenchymal and endothelial cells.	For Fgfr2-null: Day 80 For Ctnnb-null: Day 50	Mori et al., 2019 [11]
Lung	Mouse host, mouse donor	Fgf10 Ex1^mut.^/ Ex3^mut^	Histologically and morphologically normal lungs compared to wild-type, cells were a mix of GFP +ve ESC-derived cells and host cells.	4 months	Kitahara et al., 2020 [12]
Lung	Mouse host, mouse donor	Foxa2 driven fgfr2 KO	No significant difference in pulmonary function test (airway resistance, frequency, tidal volume, expiratory flow at 50% expired tidal volume) in chimeric mice.	4 weeks	Miura et al., 2023 [13]
Lung	Mouse host, mouse donor	NKx2-1 KO	Rescued lung and thyroid tissue, with ESC-derived cells expressing similar gene expression and differentiation characteristics (such as surfactant production, T1α expression).	No survival after birth	Wen et al., 2020 [14]
**Lung**	**Mouse host, rat donor**	NKx2-1 KO	Rescued lung tissue, with 30% of all Nkx2-1 homozygous knockout mouse–rat chimeras demonstrating 98.5% cell contribution from mouse ESCs. RNAseq showed normal gene expression profiles and cell signaling pathways in the chimeras.	E20.5	Wen et al., 2024 [15]
Kidney	Mouse host, mouse donor	FAH KO	Significant repopulation of proximal tubular cells, >50% donor-derived. Restored kidney function, no significant difference in creatinine levels compared to wild-type mice.	Day 300 for liver function tests and survival, no specific timeline concerning kidney function testing	Espejel et al. [7]
Kidney	Mouse host, mouse donor	Sall1 KO	Histologically and morphologically normal kidneys. Both ESC- and iPSC-complemented mice showed high contribution in kidney epithelia (except collection tubules). Stromal elements (vessels, nerves) were a mix of host and donor cells.	No survival to adulthood	Usui et al., 2012 [16]
**Kidney**	**Rat host, mouse donor**	Sall1 KO	Morphologically rescued kidneys. High mouse contribution in metanephric mesenchymal cells, but collecting tubules and blood vessels showed a mix of donor mouse and host rat cells. Successful connection between ureter and bladder. Decreased kidney size compared to control rats, but similar to wild-type mice. Size of glomeruli like control rats; number of glomeruli similar to control mice.	No survival to adulthood	Goto et al., 2019 [17]
Kidney	Pig host, pig donor	Sall1 KO	First attempt did not result in successful kidney development. Second attempt led to 1 chimera from 97 complemented blastocysts with histologically and morphologically normal kidney.	E43	Matsunari et al., 2020 [8]
**Kidney**	**Pig host, human donor**	Six1, Sall1 KO	Histologically similar mesonephros to wild-type embryos, and similar mesonephric tubule density. DsRed-labeled human-derived contribution was around 50–65% for all mesonephric cells, with over 60% in mesonephric tubules, but under 40% in mesenchyme.	E25 or E28	Wang et al., 2023 [18]
**Pancreas**	Mouse host, mouse donor **Mouse host, rat donor**	PDX1 KO	Mouse–mouse chimeras: Functionally, histologically and morphologically normal pancreas. Pancreatic islets, ducts and exocrine tissue entirely derived from mouse donor cells in PDX1^−/−^ mice. When transplanted in diabetic mice, normal serum glucose and normal response to glucose tolerance test. Mouse–rat chimeras: Pancreatic epithelia was fully composed of rat-derived cells. Of the 2 chimeras that reached full maturity, histological and morphological analysis was normal, with normal serum glucose levels and glucose tolerance testing.	PDx1^−/−^ Mouse–Mouse: 60 days post-transplantation of iPSC-derived pancreas into diabetic mice PDX1^−/−^ Mouse–Rat: 8 weeks (only 2/10 survived to adulthood)	Kobayashi et al., 2010 [19]
Pancreas	Pig host, pig donor	Introduction of Pdx1-Hes1 transgene	Histologically normal pancreas, almost all pancreatic cells derived from donor cells. Normal serum glucose levels, with 1 chimeric pig showing normal oral glucose tolerance test.	Minimum of 12 months	Matsunari et al., 2013 [20]
**Pancreas**	**Rat host, mouse donor**	Pdx1 KO	Morphologically normal pancreas, homogeneously expressing mouse-derived cells. Supporting tissue did demonstrate host rat-derived cells. Mouse-derived pancreatic islets were then re-transplanted in mice, with normal glycemic levels during follow up. Islet cells showed successful hormone secretion with insulin, glucagon and somatostatin expression.	Normal glycemic levels for up to 370 days following transplantation of pancreas derived from blastocyst complementation	Yamaguchi et al., 2017 [21]
Pancreas	Pig host, pig donor	Pdx1 KO	Histologically and morphologically normal pancreas in 2/4 chimeric animals. In pigs with successful pancreatic rescue, high levels of chimerism could be shown.	Full-term fetuses	Matsunari et al., 2020 [8]
**Heart**	Mouse host, mouse donor **Mouse host, rat donor**	Nkx2.5-Cre- and Tie2-Cre- dependent DTA (diptheria toxin A)	Mouse–mouse chimera: Donor-derived endothelial cells and cardiomyocytes, with normal functioning hearts in 8 chimeras. No signs of fibrosis. In 3 chimeras, cardiomyocyte area and vascular density comparable to control. Rat–mouse chimera: Heart complementation with almost complete donor-derived cardiomyocytes in Nkx2.5-Cre mice at E10.5. Unsuccessful heart or vascular system complementation in Nkx2.5-Cre;Tie-2Cre mice in later developmental stages (E11.5, E14.5).	Mouse–mouse chimera: up to adulthood Mouse–rat chimera: E10.5 (Nkx2.5-Cre), E10.5, E11.5, E14.5 (Nkx2.5-Cre;Tie2-Cre)	Cappiello et al., 2023 [22]
Thyroid	Mouse host, mouse donor	Fgf10 Ex1^mut^/Ex3^mut^	Morphologically and histologically normal thyroid in neonatal and adult mice. GFP expression was 86.4% +/− 7.9% in thyroid follicular cells. GFP expression did not dominate in C-cells, vasculature and connective tissue. T3 and T4 levels comparable to wild-type.	Adulthood	Ran et al., 2020 [23]
Thyroid	Mouse host, mouse donor	NKx2-1 KO	Rescued thyroid tissue in chimeric mice, with efficient donor contribution to thyrocyte progenitor cells.	E17.5	Wen et al., 2021 [14]
Thymus	Mouse host, mouse donor	Fox N1 KO	Rescued thymus in 11 mice. In 2 mice examined, 98 and 96.9% of thymic epithelial cells were donor-derived. Compared to normal mice: No significant difference in number of peripheral T cells or gene-expression profile. In splenic T cells, no significant difference in CD4+ or CD8+ T cell proliferation or production of IFNγ, IL-2 and Granzyme B with anti-CD3 stimulation. Under anti-PDL1 treatment: suppression of MC38 tumor growth and increased IFNγ production and T cell activation (via decrease in PD1 expression).	Up to 42 weeks	Yamazaki et al., 2022 [24]
Thymus	Mouse host, mouse donor	Foxa2-driven Fgfr2 KO	Rescued thymic phenotype. Chimerism in thymus: average 92.4% (SD 5.1) in thymic epithelium; average 52.9% (SD 20) in thymic mesenchyme.	Up to 4 weeks	Miura et al., 2023 [13]
**Parathyroid**	Mouse host, mouse donor **Rat host, mouse donor**	GCM2 KO	Mouse–mouse: Histologically normal parathyroids. GFP donor-derived signal was 94.6% in chief cells, 65.2% in endothelial cells and 45.6% in mesenchymal cells. Function: compared to control mice, similar plasma calcium levels, basal PTH levels and PTH stimulation response. Gene expression level: compared to control mice, increased GATA3 and GCM2, and similar levels of Mafb, Casr and PTH. Rat–mouse: rescued parathyroid phenotype, successful expression of transcription factors necessary for further development and PTH.	Mouse–mouse: adulthood Rat–mouse: death soon after birth	Kano et al., 2023 [25]

## Data Availability

No new data were created or analyzed in this study.

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
