# Peer review of "Interspecies Blastocyst Complementation and the Genesis of Chimeric Solid Human Organs"

_genes, 2025, doi:10.3390/genes16020215_

Round 1
Reviewer 1 Report
Comments and Suggestions for Authors
This manuscript examines recent progress in blastocyst complementation for the generation of solid organs. Although the content addresses a compelling subject, we have outlined numerous issues and suggestions for enhancement:
Major considerations:
- Liver: the study from S.Espejel et al., (doi: 10.1172/JCI43267) assessing the interspecies complementation in FAH-deficient blastocysts lacks in the list of studies and is not discussed
- Table1:
- I would suggest to add survival rates (%) in the survival column.
- I would suggest to highlight the interspecies complementation in the table
- Standardize the time units (e.g., use months/weeks/days consistently instead of a mix of days, weeks, and months). For example, specify "21 days after birth" where applicable. If there is no survival, simply state "No survival."
- Reorder "pig" and "mice" in row 2 to match the order in the "Species" column, starting with "pig."
- Line 96: Based on the latest reference (2016), were there any clinical improvements reported? If available, include recent studies with more mature hepatocyte data and corresponding survival rates.
- Line 630: There have been numerous studies investigating interspecies blastocyst complementation using non-deficient blastocysts. I understand that these were outside the scope of this review, but it would be interesting to mention some of them to discuss the role of the "type" of donor stem cell that is most likely to succeed in complementation (iPSC, naive ES, primed ES...).
- Line 645: I would suggest introducing an entire paragraph about the development of vascularization/endothelial cells. In the field of solid organ transplantation, this would be crucial for the development of the technology in the clinical settings. In this domain, the studies using the Flk1-deficient blastocysts and ETV2-deficient blastocysts should be cited/discussed (https://doi.org/10.1016/j.stemcr.2018.08.015 and doi: 10.1038/s41587-019-0373-y).
- Line 661: Can you clarify how inhibition of apoptosis by BMI1 and BCL2 overexpression is associated with the increased efficacy of human contributions in mice?
Minor considerations:
1- The manuscript requires a thorough review to ensure uniformity in abbreviation usage for ex: use ESC instead of “embryonic stem cells”, enhanced transitions between subjects, and the incorporation of further details to support the presented data.
2- Define each abbreviation at its first occurrence, e.g., in line 60, introduce "RAG2."
3- Use a uniform pattern for introducing abbreviations, e.g., "producing parathyroid hormone (PTH)" (line 580) instead of "DTA (diphtheria toxin subunit A)" (line 489).
4- Line 81-84: Consider rewording the table title/legend for clarity. Suggested revision: “The summary of results from all 22 studies, including details on the organs generated, the host and donor species, the cells used for complementation, the target gene modifications (such as knock-out [KO] or gene introduction) to achieve organ agenesis, the functionality of the de novo organs, and the survival or follow-up timelines in chimeric animals”.
5- Line 86: Clarify the meaning of "CDC."
6- Line 141: This sentence is somewhat vague; can you add details for better clarification? for example you can mention the total number of complemented embryos.
7- Line 149: Enhance the sentence by adding "to minimize off-target effects" after "interest".
8- Line 164: Clarify the meaning of “COPD”.
9- Lines 184-196: Needs revision, add more details and improve transitions between sentences.
10- Lines 198-216: Address redundancies to enhance readability and consistency in gene nomenclature (e.g., use either "fgfr" or "Fgfr" throughout, as per journal guidelines).
11- Line 234: Replace “system” with organs.
12- Line 320: “Moreover, they did not have success generating kidneys when injecting rat iPSCs into Sall1-/- mice”, is better to be replaced with “Moreover, they were unsuccessful in generating kidneys when injecting rat iPSCs into Sall1-/- mice”.
13- Line 336: Is Sall1-/0 correctly represented?
14- Lines 346 to 348: Provide evidence or references supporting potential contributions to brain tissue.
15- Line 356: Update data if newer information is available.
16- Line 373: "resembling that of adult pancreases" needs to be supported by more details.
17- Line 377: Replace "differentiation" with "differentiating."
18- Line 403: Suggested replacement: “and may achieve increased overall functionality”.
19- Line 468: Define BMP by its first mention.
20- Line 676: Avoid repeating abbreviations already introduced, e.g., "mESCs." Apply this throughout the manuscript.
21- Line 706 and further: the single cell RNA sequencing is an interesting technology but there are redundancies between the lines 706-715 and the next paragraph.
22- Line 740 and 742: Avoid redundancy: Replication timing (RT).
Author Response
Major considerations:
- Liver: the study from S.Espejel et al., (doi: 10.1172/JCI43267) assessing the interspecies complementation in FAH-deficient blastocysts lacks in the list of studies and is not discussed
Thank you for this suggestion, we agree this is an important study to include. The study was added in the table, as well as in the text under both liver and kidney (Starting lines 139 and 335, respectively).
- Table1:
- I would suggest to add survival rates (%) in the survival column.
While we agree this would be an important aspect to add, not all studies comprehensively included exact follow up, particularly the number of animals that survived to the specified time points in the survival column.
- I would suggest to highlight the interspecies complementation in the table
Studies that included interspecies complementation have now been highlighted in bold in the table of the revised manuscript.
- Standardize the time units (e.g., use months/weeks/days consistently instead of a mix of days, weeks, and months). For example, specify "21 days after birth" where applicable. If there is no survival, simply state "No survival."
Some papers did not explicitly state exact survival time, with only ‘adulthood’ being specified. It is difficult to standardize time units, as timelines were very different between studies. Furthermore, specifying whether full gestation was achieved, or only survival to day 20 of development would be a significant difference. Therefore, stating “No survival past birth” could conceivably over simplify the timeline.
- Reorder "pig" and "mice" in row 2 to match the order in the "Species" column, starting with "pig."
Thank you for the suggestion and the order is now corrected in the revised manuscript to match that which is in the species column.
- Line 96: Based on the latest reference (2016), were there any clinical improvements reported? If available, include recent studies with more mature hepatocyte data and corresponding survival rates.
A more recent study was included, detailing survival rates and markers of hepatic rescue (Takayama et al, 10.1002/hep4.1111). Starting line 101
- Line 630: There have been numerous studies investigating interspecies blastocyst complementation using non-deficient blastocysts. I understand that these were outside the scope of this review, but it would be interesting to mention some of them to discuss the role of the "type" of donor stem cell that is most likely to succeed in complementation (iPSC, naive ES, primed ES...).
We agree this is an important aspect to outline, and therefore a paragraph has been added in the “Challenges…” section of the revised manuscript. This allows a discussion of the importance of different type of donor cells used to achieve more successful complementation. (starting line 731)
- Line 645: I would suggest introducing an entire paragraph about the development of vascularization/endothelial cells. In the field of solid organ transplantation, this would be crucial for the development of the technology in the clinical settings. In this domain, the studies using the Flk1-deficient blastocysts and ETV2-deficient blastocysts should be cited/discussed (https://doi.org/10.1016/j.stemcr.2018.08.015 and doi: 10.1038/s41587-019-0373-y).
A paragraph has been added to the revised manuscript (starting line 703) discussing the use of blastocyst complementation for the generation of endothelial/vascular cells using the above mentioned references.
- Line 661: Can you clarify how inhibition of apoptosis by BMI1 and BCL2 overexpression is associated with the increased efficacy of human contributions in mice?
Further clarification was included at the end of the paragraph (starting line 725)
Minor considerations:
- The manuscript requires a thorough review to ensure uniformity in abbreviation usage for ex: use ESC instead of “embryonic stem cells”, enhanced transitions between subjects, and the incorporation of further details to support the presented data.
- Abbreviations uniformly corrected (particularly for ESC and iPSC)
- Define each abbreviation at its first occurrence, e.g., in line 60, introduce "RAG2."
- Each abbreviation has now been defined at first occurrence.
- Use a uniform pattern for introducing abbreviations, e.g., "producing parathyroid hormone (PTH)" (line 580) instead of "DTA (diphtheria toxin subunit A)" (line 489).
- This has been corrected accordingly.
- Line 81-84:Consider rewording the table title/legend for clarity. Suggested revision: “The summary of results from all 22 studies, including details on the organs generated, the host and donor species, the cells used for complementation, the target gene modifications (such as knock-out [KO] or gene introduction) to achieve organ agenesis, the functionality of the de novo organs, and the survival or follow-up timelines in chimeric animals”.
We thank the reviewer for this suggestion. The table legend in the revised manuscript has been reworded accordingly.
- Line 86: Clarify the meaning of "CDC."
- The CDC abbreviation has been clarified and included in the text.
- Line 141: This sentence is somewhat vague; can you add details for better clarification? for example you can mention the total number of complemented embryos.
The sentence was clarified to include the number of embryos recovered from total complemented embryos in the revised manuscript. (Line 162)
7- Line 149: Enhance the sentence by adding "to minimize off-target effects" after "interest".
The suggested wording was added to the sentence.
8- Line 164: Clarify the meaning of “COPD”.
The abbreviation for COPD was clarified and added in the text.
9- Lines 184-196: Needs revision, add more details and improve transitions between sentences.
We agree with the reviewer. This section has now been edited to include greater detail, particularly for Fgf10, and improved transitions between factors as recommended. (starting line 207)
10- Lines 198-216: Address redundancies to enhance readability and consistency in gene nomenclature (e.g., use either "fgfr" or "Fgfr" throughout, as per journal guidelines).
Gene nomenclature was edited for consistency during the text. It was corrected to Fgfr or Fgf.
11- Line 234: Replace “system” with organs.
This has been corrected accordingly.
12- Line 320: “Moreover, they did not have success generating kidneys when injecting rat iPSCs into Sall1-/- mice”, is better to be replaced with “Moreover, they were unsuccessful in generating kidneys when injecting rat iPSCs into Sall1-/- mice”.
This has been corrected accordingly in the revised manuscript.
.13- Line 336: Is Sall1-/0 correctly represented?
This should state -/-, and has been corrected accordingly.
14- Lines 346 to 348: Provide evidence or references supporting potential contributions to brain tissue.
We have now included a reference in support of potential contributions to brain tissue, and particularly in terms of brain development when age-matched to humans. Line 386. Ref: Stiles et al, doi:10.1007/s11065-010-9148-4
15- Line 356: Update data if newer information is available.
We have not been able to find more recent data for kidney blastocyst complementation, beyond the study mentioned in 2023.
16- Line 373: "resembling that of adult pancreases" needs to be supported by more details.
We have included additional details, with reference to specify the exact proportions of cells in adult pancreases. Line 413, Ref: Ionescu-Tirgoviste et al, DOI: 10.1038/srep14634
17- Line 377: Replace "differentiation" with "differentiating."
This has been corrected accordingly.
18- Line 403: Suggested replacement: “and may achieve increased overall functionality”.
Revised according to the above mentioned suggestion.
19- Line 468: Define BMP by its first mention.
BMP has now been defined at its first mention.
20- Line 676: Avoid repeating abbreviations already introduced, e.g., "mESCs." Apply this throughout the manuscript.
This has been corrected accordingly and is consistent throughout the text of the revised manuscript.
21- Line 706 and further: the single cell RNA sequencing is an interesting technology but there are redundancies between the lines 706-715 and the next paragraph.
The paragraph immediately following section 9.2.1 has been edited, shortened and merged into the following paragraph to avoid redundancies (Starting line 800)
22- Line 740 and 742: Avoid redundancy: Replication timing (RT).
This has been corrected; and we appreciate the reviewer’s careful and detailed review of the manuscript.
Reviewer 2 Report
Comments and Suggestions for Authors
Review of the manuscript
„Interspecies Blastocyst Complementation and the Genesis of 2
Chimeric Solid Human Organs"
Elena Bigliardi and colleagues present a comprehensive review of 22 studies describing the generation of solid organs using blastocyst complementation. The experiments are contextualized within the broader goal of creating human organs to address the worldwide shortage of human donor organs for transplantation. The authors highlight recent advancements and effectively explain the challenges associated with this technique. Furthermore, they provide an outlook, concluding that significant hurdles remain to be overcome in creating human organs using blastocyst complementation.
Comments:
The authors present a very detailed manuscript, with sections dedicated to each solid organ that may be created using blastocyst complementation. The omission of tissue such as skeletal muscle is a welcome aspect of the review. The quality of the text is exceptionally high, and all explanations and conclusions are sound. In my opinion, this is a very good and helpful review. Nevertheless, I have some suggestions for the authors.
1. Consider consolidating information on stem cells into a separate paragraph:
In addition to the proper niche for the transferred stem cells, the quality and abilities of the stem cells used for complementation determine the features of the organ that will fill the niche. In contrast to well-established murine and human embryonic or induced pluripotent stem cells, pig stem cells were previously unable to be cultured in vitro for an extended period. However, recent advancements have improved this situation, and this development should be mentioned in the review. I suggest that the authors consolidate the information regarding stem cells used for complementation from different species into a separate paragraph.
2. Add information to ethical aspects:
While the potential threat of creating human neural tissue in chimeric animals is mentioned, many other unresolved issues will arise once blastocyst complementation is successful. These include legal issues, such as the ownership of created organs, their distribution, and the applicable regulations. These points could be incorporated into the review.
3. Shorten the outlook
The outlook should be focused on the most critical points that need to be addressed. These points do not require detailed explanation, and therefore the paragraph should be shortened.
Summary:
This is a very good review that needs a minor revision.
Author Response
- Consider consolidating information on stem cells into a separate paragraph:
In addition to the proper niche for the transferred stem cells, the quality and abilities of the stem cells used for complementation determine the features of the organ that will fill the niche. In contrast to well-established murine and human embryonic or induced pluripotent stem cells, pig stem cells were previously unable to be cultured in vitro for an extended period. However, recent advancements have improved this situation, and this development should be mentioned in the review. I suggest that the authors consolidate the information regarding stem cells used for complementation from different species into a separate paragraph.
Thank you for this suggestion; and we agree this is a very important aspect to discuss. We have now included a paragraph discussing the different stem cell types that may be used for complementation, relevant to the interspecies differences that exist. (starting line 731)
- Add information to ethical aspects:
While the potential threat of creating human neural tissue in chimeric animals is mentioned, many other unresolved issues will arise once blastocyst complementation is successful. These include legal issues, such as the ownership of created organs, their distribution, and the applicable regulations. These points could be incorporated into the review.
We thank the reviewer for this suggestion and agree that these points should be included in the review. In the revised manuscript, we have added further discussion of the legal and ethical implications of blastocyst complementation (starting line 863)
- Shorten the outlook
The outlook should be focused on the most critical points that need to be addressed. These points do not require detailed explanation, and therefore the paragraph should be shortened.
We have modified the outlook and separated the sections into challenges (section 9) and conclusion (section 10) to shorten and only discuss the most critical points that need to be addressed.
Summary:
This is a very good review that needs a minor revision.
The authors would like to thank Reviewer 2 for the recommendation that the review is very good and needs only minor revision. We hope that our manuscript is now acceptable for publication in Genes.